# Two-Stage Neural Contextual Bandits for Adaptive Personalised Recommendations

## Abstract

We consider the problem of personalised recommendations where each user consumes recommendations in a sequential fashion. Personalised recommendation methods that focus on exploiting user interests but ignore exploration will result in biased feedback loops, which hurt recommendation quality in the long term. In this paper, we consider contextual bandits based strategies to address the exploitation-exploration trade-off for large-scale adaptive personalised recommendation systems. In a large-scale system where the number of items is exponentially large, addressing the exploitation-exploration trade-off becomes significantly more challenging that renders most existing standard contextual bandit algorithms inefficient. To systematically address this challenge, we propose a hierarchical neural contextual bandit framework to efficiently learn user preferences. Our hierarchical structure first explores dynamic topics before recommending a set of items. We leverage neural networks to learn non-linear representations of users and items, and use upper confidence bounds (UCBs) as the basis for item recommendation. We propose an additive linear and a bilinear structure for UCB, where the former captures the representation uncertainties of users and items separately while the latter additionally captures the uncertainty of the user-item interaction. We show that our hierarchical framework with our proposed bandit policies exhibits strong computational and performance advantages compared to many standard bandit baselines on two large-scale standard recommendation benchmark datasets.

## 1 Introduction

Online platforms rely on effective and efficient user modelling and personalised recommendation (Wu et al., 2021a). The recommender system faces the *exploitation-exploration dilemma*, where one can exploit by recommending items that the users like the most so far, or one can also explore by recommending items that users have not browsed before but may potentially like (Li et al., 2010). Focusing on exploitation in the early learning phase tends to create a pernicious feedback loop, which amplifies biases and raises the so-called *filter bubbles* or *echo chamber* (Jiang et al., 2019), where the exposure of items is narrowed by a self-reinforcing pattern.

Contextual bandits are designed to address the exploitation and exploration dilemma and have been used to mitigate the feedback loop effect (Chen et al., 2020; Li et al., 2010) by user interest and item popularity exploration. One can formalise the online recommendation problem as sequential decision-making under uncertainty, where given some contextual information, an agent (the recommender system) selects one or more arms (the items) from all possible choices according to a policy (recommendation strategy), with the goal of designing a policy which maximises the cumulative rewards (user clicks).

A large scale commercial recommender system has millions of dynamically generated items. Calculating the acquisition scores for all candidate items can be computationally expensive (Li et al., 2010). To improve the computational efficiency, we propose a hierarchical topic-item model (Algorithm 1), where for a given user, we select topics first and then recommend items from the selected topics. Recent work considering hierarchical searches constrained to pre-constructed tree structures and shallow representations (Wang et al., 2018; Song et al., 2021). In contrast, we dynamically construct topic groups using bandit acquisition scores,

which allows a more efficient topic-level exploration and addresses the potentially imbalanced topic size (Algorithm 2).

State-of-the-art recommender systems make use of deep neural networks (DNN) to learn user and item representations (Wu et al., 2021b; 2019). How to make use of the power of deep representation and calculate uncertainties (i.e., a confidence interval for predictions) is the key point of efficient exploration. To enable the current industrial pipeline to be extended to further explore user interest without large deployment costs, the best approach is to retain the current DNN structures deployed in the system and add the modules to infer uncertainties. Achieving efficient uncertainty inference and sufficiently accurate estimates at the same time with existing DNN structures is challenging. For example, Guo et al. (2020); Duran-Martin et al. (2022) considered Bayesian DNN, which is computationally expensive to maintain Bayesian neural models and updates for large-scale systems. Riquelme et al. (2018) provided baselines on how to infer uncertainties with existing deep models, however, without sufficient accuracy and large item space exploration efficiency.

Our proposed new bandit policies in Section 3.3, namely the Shared Neural Generalised Additive and Bilinear Upper Confidence Bound (S-N-GALM-UCB, S-N-GBLM-UCB) policies, address the above challenge and *can be plugged into any existing DNN structures* with an additional generalised linear layer to infer uncertainties and shows improvement over Riquelme et al. (2018). We consider sharing parameters with all users/items to use samples efficiently and generate uncertainties. S-N-GALM-UCB has relatively small parameter space and is designed to efficiently capture user and item uncertainties separately. S-N-GBLM-UCB suits the two-tower structure naturally and can capture the uncertainty of user-item interaction. It has better performance for dense interactions but may take longer running time without low-rank approximation.

Our goal is to serve existing NN-based recommendation pipelines and add exploration to alleviate feedback loop bias. We consider a contextual bandits framework to alleviate the feedback loop bias and address challenges of computational efficiency with large item space and neural uncertainty inference, which is illustrated in Figure 1. The proposed two parts are naturally combined and serve our overall goal together: shared parameters and additive/bilinear structure in bandit policies help the dynamic topics to be formed efficiently with good quality. In return, hierarchical design reduces the exploration space for bandit policies. We evaluate our proposed framework empirically on two real-world large-scale datasets [1], namely a news recommendation dataset MIND (Wu et al., 2020) and a movie recommendation dataset MovieLens-20M (Harper & Konstan, 2015). Empirical results show our proposed new bandit policies outperform baseline algorithms (Table 4) and our two-stage design is more computationally efficient than one-stage algorithms (Table 5). Our contributions are:

- We propose a hierarchical two-stage topic-item neural contextual bandit framework (Algorithm 1) for user interest exploration in the recommender system, where topics and items are considered as arms in the first and second stage, respectively.

- To address the topic imbalance problem, we propose to construct topics dynamically to balance the number of items per topic (Algorithm 2).

- We propose two bandit policies for recommendation: shared neural generalised additive, and bilinear upper confidence bound, that use deep representation of contextual information efficiently.

- We show our proposed methods have a strong benefit in terms of both computational efficiency and empirical performance on two large-scale real-world recommendation datasets.

## 2 Problem Setting

**Personalised Recommender System** We consider a recommender system that sequentially recommends personalised items to users, with the goal of maximising cumulative clicks for all users. The recommender system learns from the interaction history with the users, and for any given user, the system displays several items selected from the candidate item set. Then the user will react as either click or non-click and the system uses this as feedback to learn the preference of users. This task is challenging since the candidate

---

[1]Source code is available at `https://anonymous.4open.science/r/CB4Rec-8779/`

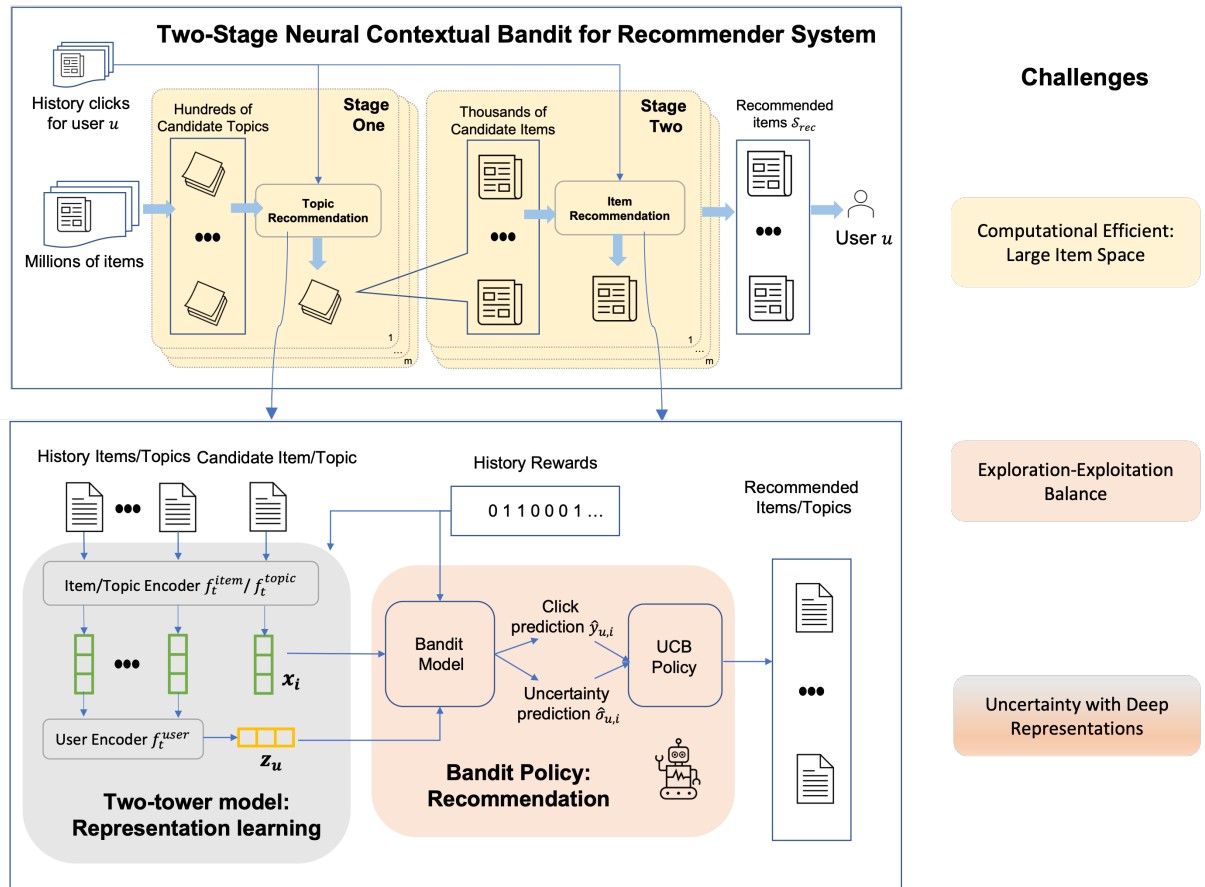

Figure 1: Two-stage topic-item neural contextual bandit framework for recommender system. Topic and item recommendation follows the same strategy style shown in the bottom box. We address the large item space using a two-stage hierarchy of topics and items. We propose two generalisations of contextual bandits policies to balance exploration-exploitation and a dynamic topic construction to address the imbalanced topic problem with two-stage recommendations. A two-tower model (user and item encoder) is used to learn deep representations of users and items. Our framework can be generalised to any model which can generate item/topic and user representations.

item set is in the order of millions and dynamically changes over time. In addition, there are a large number of cold users (i.e., users that do not have any history) and the user interest can shift over time (Wu et al., 2021a). This problem can be formulated as a sequential decision-making problem, studied in the field of contextual bandits (Li et al., 2010; Song et al., 2021).

**Problem Setup**   We first introduce the general contextual bandit problem formulation. A recommender system is regarded as an *agent*, items are *arms* (choices), and the user and item representation form the *context*. At each iteration $t = 1, \ldots, N$, given user $u_t$ and candidate arm set $\mathcal{A}_t$, one can generate the item representation $\boldsymbol{x}_i \in \mathbb{R}^{d_1}$ for all $i \in \mathcal{A}_t$, and the user representation $\boldsymbol{z}_{u_t} \in \mathbb{R}^{d_2}$ as context. In the following, we will drop subscript $t$ for $u_t$ when there is no ambiguity. The agent recommends $m \geq 1$ items, denoted as $\mathcal{S}_{rec}$, according to a *policy* $\pi$ given the context. Then the agent receives the feedback $\{y_{t,1}, \ldots, y_{t,m}\}$, where $y_{t,i} \in \{0, 1\}$ indicating whether the user clicks the item $i$ or not at iteration $t$. The *reward* is defined as $y_t = \sum_{i=1}^m \mathbb{I}\{y_{t,i} = 1\}$, where $\mathbb{I}\{\cdot\}$ is the indicator function. The goal is to design a policy to minimise the expected cumulative regret (Definition 1), which is equivalent to maximising the expected cumulative rewards (Li et al., 2010; Song et al., 2021). Since in recommender systems the optimal rewards are usually unknown, we focus on maximising the cumulative rewards in this work. We summarise our main notations in Table 1 in Appendix.

**Definition 1.** *For a total iteration $N$, the expected cumulative rewards are $\mathbb{E}\left[\sum_{t=1}^N y_t\right]$, where the expectation is taken over the randomness of the context, rewards and policy. Let the optimal reward for user $u_t$ as $y_t^*$, the expected cumulative regret is defined as $\mathbb{E}\left[\sum_{t=1}^N (y_t^* - y_t)\right]$.*

**Bandits Policies**   Upper Confidence Bound (UCB) is one type of classical bandit policy proposed to address the exploration-exploitation dilemma and proven to have sublinear regret bound (Auer et al., 2002). The idea is to pick the arm with the highest UCB *acquisition score*, which captures the upper confidence bound for predictions in high probability. The generic form of the UCB *acquisition function* for a user-item pair $(u, i)$ is as follows

$$\alpha^{UCB}(u, i) := \hat{y}_{u,i} + \beta \hat{\sigma}_{u,i}, \tag{1}$$

where $\hat{y}_{u,i}$ is the click prediction, $\hat{\sigma}_{u,i}$ is the uncertainty of predictions, and $\beta$ is a hyperparameter balancing the exploitation and exploration. Li et al. (2010) popularised the LinUCB contextual bandits approach on news recommendation tasks, where the expected reward of user $u$ and item $i$ is assumed to be linear in terms of the contextual feature $\boldsymbol{c}_{u,i} \in \mathbb{R}^d$. Xu et al. (2020) and Riquelme et al. (2018) studied neural linear models with representation of contextual information learnt by neural networks, which further improves the performance. Filippi et al. (2010) extended the LinUCB policy to the Generalised Linear Model s.t. $\mathbb{E}[y_{u,i}|\boldsymbol{c}_{u,i}] = \rho(\boldsymbol{c}_{u,i}^T \boldsymbol{\theta}_u^*)$, where $\rho : \mathbb{R} \to \mathbb{R}$ is the inverse link function, $\boldsymbol{\theta}_u^* \in \mathbb{R}^d$ is the unknown coefficient. When $\rho(\boldsymbol{x}) = \boldsymbol{x}$, the problem is reduced to linear bandits. Define the *design matrix* $D_u \in \mathbb{R}^{n_u \times d}$ at iteration $t$, where each row contains sample interacted with user $u$. With $M_u = D_u^T D_u + \boldsymbol{I}_d$ and estimated coefficient $\hat{\boldsymbol{\theta}}_u$, the GLM-UCB acquisition function follows

$$\alpha^{GLM-UCB}(u, i) := \rho(\boldsymbol{c}_{u,i}^T \hat{\boldsymbol{\theta}}_u) + \beta \|\boldsymbol{c}_{u,i}\|_{M_u^{-1}}, \tag{2}$$

where $\|\boldsymbol{c}_{u,i}\|_{M_u^{-1}} = \sqrt{\boldsymbol{c}_{u,i}^T M_u^{-1} \boldsymbol{c}_{u,i}}$. In this paper, we consider GLM-UCB policy as a base policy. Since user feedback is binary, we use the sigmoid function, i.e., we set $\rho(x) = \exp(x)/(1 + \exp(x))$, which is the inverse link function of a Bernoulli distribution (Nelder and Wedderburn, 1972).

## 3   Methods

In this section, we introduce our proposed methods for the large-scale personalised recommender problem and we address the challenges shown in Section 1. We first propose a two-stage exploration framework in Section 3.1 and then describe our proposed bandit policies in Section 3.3.

| Notation | Description |
|---|---|
| $N$ | Number of iterations (i.e. interaction rounds) |
| $T$ | Number of trials |
| $m$ | Number of recommendations per iteration |
| $\mathcal{A}_t$ | Candidate arm set at iteration $t$ |
| $\boldsymbol{x}$ | item embedding |
| $\boldsymbol{z}$ | user embedding |
| $\mathcal{S}_{rec}$ | Recommendation set |
| $y_{u,i}$ | (Binary) Preference of user $u$ for item $i$ |
| $\hat{y}_{u,i}$ | Predicted preference of user $u$ for item $i$ |
| $\hat{\sigma}_{u,i}$ | Predicted preference uncertainty of user $u$ for item $i$ |
| $y_t$ | Reward at iteration $t$ |
| $y_t^*$ | Optimal reward at iteration $t$ |
| $\alpha(u,i)$ | Acquisition score for user $u$ and item $i$ |
| $\beta$ | Balancing hyper-parameter for exploitation-exploration |
| $\rho$ | inverse link function |
| $p$ | Dynamic topic minimum reconstruct size |
| $B$ | Computation budget for each iteration |

Table 1: A collection of our notations used in the paper.

---

**Algorithm 1** Two-stage Exploration Framework

---

1: **Input:** recommendation size $m$, # iterations $N$, # topics $q$, user set $\mathcal{U}$, topic set $\mathcal{V}$, item set $\mathcal{S}_{v_j}$ associated with topic $v_j$, topic and item acquisition functions $\alpha_1$, $\alpha_2$, resp., minimum reconstruction topic size $p$
2: **for** $t = 1$ to $N$ **do**
3:     Random user $u_t \sim \text{Unif}(\mathcal{U})$ emerges to the recommender
4:     **Stage One: Topic recommendation**
5:     Compute topic scores $\alpha_1(u_t, v_j)$, $\forall v_j \in \mathcal{V}$
6:     Obtain $(v_{(1)}, \ldots, v_{(q)})$ s.t. $\alpha_1(u_t, v_{(1)}) \geq \cdots \geq \alpha_1(u_t, v_{(q)})$
7:     Pick sets of topics as $\mathcal{S}^1 = \{v_{(1)}\}, \ldots, \mathcal{S}^m = \{v_{(m)}\}$
8:     Obtain dynamically reconstructed topic group arms $\{\mathcal{S}^1, \ldots, \mathcal{S}^m\} \leftarrow$ DynaTopicRec$(m, (v_{(1)}, \ldots, v_{(q)}), p)$ (Alg. 2)
9:     **Stage Two: Item recommendation**
10:     Initialise recommendation set $\mathcal{S}_{rec} = \{\}$
11:     **for** j = 1 to m **do**
12:         Compute item scores $\alpha_2(u_t, i)$, $\forall i \in \mathcal{S}_v$, $\forall v \in \mathcal{S}^j$.
13:         Add item $i^*$ with the highest score $\alpha_2(u_t, i^*)$ to $\mathcal{S}_{rec}$.
14:     **end for**
15:     Get rewards $[y_{u_t,a}]_{a \in \mathcal{S}_{rec}} = \text{Oracle}(u_t, \mathcal{S}_{rec})$
16:     Update the topic and item models with $\{(a, y_{u_t,a})\}_{a \in \mathcal{S}_{rec}}$
17: **end for**

---

### 3.1 Two-Stage Topic-Item Exploration Framework

Recall our goal is to sequentially recommend $m \geq 1$ items to users in a large scale recommender system. To reduce the computational complexity of whole-space item exploration, we consider a two-stage exploration framework in Algorithm 1. We call each of the $m$ recommendations as *recommendation slot*. In stage one (line 5-8), we recommend a set of topics for each recommendation slot. Each topic is treated as an arm, and we decide which topics can be recommended by the topic acquisition function $\alpha_1$. For example, one can use the UCB acquisition function defined in Eq. (1) or (2) as $\alpha_1$. For each recommendation slot, we initialise the set of recommended topics with the top $m$ acquisition scores respectively. Then in line 8, we

---

**Algorithm 2** DynaTopicRec($m, (v_{(1)}, \ldots, v_{(q)}), p$) - Dynamic Topic Reconstruction

---

1: **Input:** Number of topic group arms $m \in \mathbb{N}$, sorted topics $l_{st} := (v_{(1)}, \ldots, v_{(q)})$ where $q > m$, item set $\mathcal{S}_{v_j}$ associated with topic $v_j$, minimum reconstruction topic size $p$
2: Initialise $m$ topic group arms $\mathcal{S}^i = \{v_{(i)}\}, i \in \{1, \ldots, m\}$.
3: Initialise the number of items in each topic group $c_i = |\mathcal{S}_{v_{(i)}}|, i \in \{1, \ldots, m\}$.
4: **while** there $\exists c_i \leq p, i \in \{1, \ldots, m\}$ and $|l_{st}| > 0$ **do**
5:     **for** $i = 1$ to $m$ **do**
6:         **if** $c_i \leq p$ **then**
7:             $\mathcal{S}^i \leftarrow \mathcal{S}^i \cup l_{st}[0]$
8:             $l_{st} \leftarrow l_{st} \setminus l_{st}[0]$
9:             $c_i \leftarrow c_i + |\mathcal{S}_{l_{st}[0]}|$
10:         **end if**
11:     **end for**
12: **end while**
13: **Return**: Reconstructed topic group arms $\{\mathcal{S}^1, \ldots, \mathcal{S}^m\}$.

---

dynamically expand each of the topic sets with the remaining high-score topics. In stage two (line 11-14), we select the most promising item (according to the bandit acquisition function $\alpha_2$) for each of the expanded set of topics chosen in stage one. The acquisition functions in Algorithm 1 used to recommend topics and items can follow any contextual bandits policies.

Once the agent collects $m$ recommended items (one item for each of $m$ topics), those $m$ items will be shown to the user and the agent will get user feedback, which is $m$ binary scores indicating click or non-click for each recommended item. The parameters of the topic and item neural model are updated according to the feedback every $l_t$ and $l_n$ (pre-defined hyperparameters) iterations respectively. The parameters of generalised linear models are updated at every iteration. The user representation is updated each iteration. While the hierarchical exploration framework has been used in the literature, our novelty comes from the novel design of dynamic topic set reconstruction in Section 3.2 and efficient bandit policies in Section 3.3.

### 3.2 Dynamic Topic Set Reconstruction

If the topics are highly balanced (in the sense that the number of items within each topic is roughly the same), the two-stage exploration effectively reduces the search space complexity from $\mathcal{O}(N_{\text{item}})$ into $\mathcal{O}(N_{\text{topic}} + \frac{N_{\text{item}}}{N_{\text{topic}}})$, where $N_{\text{item}}$, $N_{\text{topic}}$, and $\frac{N_{\text{item}}}{N_{\text{topic}}}$ are the total number of items, topics, and items per topic, respectively. This has a strong computational advantage in large-scale recommendations due to the large $N_{\text{item}}$ in practice. For example, in the case of the MIND dataset, as presented in Table 2, the search space complexity is reduced by about 189 times.

However, in many applications, topics are highly imbalanced, e.g. it ranges from size 1 up to 15,000 (number of items per topic) in our applications. Large topics tend to have high click predictions or low uncertain predictions, and thus are more likely to be chosen. This will increase the size of arm space in the second stage and lead to computational inefficiency. We propose to address the imbalanced topics issue by dynamically reconstructing the set of topics corresponding to each arm according to topic acquisition scores in each iteration. The main idea of forming topic sets is to include the topics with high bandits acquisition scores, which means these topics are either potential good exploitation or exploration for user interest. Furthermore, we also want to allocate topics with high acquisition scores into different topic sets, so that topics with high scores will have more chance to be selected. We initialise each topic set with the top $m$ scoring topics $\{\mathcal{S}^1, \ldots, \mathcal{S}^m\}$. Then until all topic sets have at least $p$ items, we add the topic with the highest topic acquisition score in the remaining topics to each of the $m$ topic sets in sequential order. We show the detailed description in Algorithm 2.

### 3.3 Bandit Policies: Additive and Bilinear UCB

We aim to design bandit policies that can incorporate the neural representation and inference of valid uncertainties efficiently. We consider a state-of-the-art two-tower neural network model. In Stage One, we maintain two modules at time step $t$: 1) The topic encoder $f_t^{topic}$, which takes topic information in (e.g., topic id, name) and outputs a topic representation $\boldsymbol{v} \in \mathbb{R}^{d_1}$, and 2) The user encoder $f_t^{user-topic}$, which takes the logged topic representation for user $i$ in and outputs a user representation $\boldsymbol{z}^{topic} \in \mathbb{R}^{d_2}$. In Stage Two, similarly we maintain item and user encoder $f_t^{item}, f_t^{user-item}$ and outputs item and user representation $\boldsymbol{x}, \boldsymbol{z}^{item}$. See Section 4 for our detailed neural model setup. The user and topic/item representation are treated as context, and the arms are the candidate items available to be recommended to the user.

Applying existing neural contextual bandit algorithms directly to recommender systems may be computationally expensive or lead to suboptimal performance. For example, uncertainties inferred from Monte-Carlo can have high variance (Riquelme et al., 2018); learning coefficients for each arm in neural-linear models (Riquelme et al., 2018; Zhou et al., 2020) can be unrealistic for large item space, since one needs enough samples for each of the millions of items. In our simulation, the number of users is much smaller than the items, hence we learn coefficients per user for baseline policies (e.g., GLM-UCB). From our experiment in Table 4, we observe that performance still drops when the number of users increases.

We consider *shared* bandit models where the parameters are shared by all pairs of users and (or) items/topics. Coefficient sharing across entities can make the model learning process more efficient and more generalisable. We propose two ways to capture both the user and item/topic representation in the contextual information, namely the *generalised additive linear* or *generalised bilinear* policies. Recall $\boldsymbol{x}_i \in \mathbb{R}^{d_1}$ as item $i$ representation and $\boldsymbol{z}_u \in \mathbb{R}^{d_2}$ as user $u$ representation. We use item recommendation as an example to illustrate our proposed policies below.

**Shared Neural Generalised Additive Linear UCB (S-N-GALM-UCB)** We consider an additive linear model, where the item-related coefficient $\boldsymbol{\theta}_x^*$ and user-related coefficient $\boldsymbol{\theta}_z^*$ are modelled separately, i.e., $\mathbb{E}[y_{u,i}|\boldsymbol{x}_i, \boldsymbol{z}_u] = \rho(\gamma \boldsymbol{x}_i^T \boldsymbol{\theta}_x^* + \tilde{\gamma} \boldsymbol{\theta}_z^{*T} \boldsymbol{z}_u)$, where $\gamma$ is a hyperparameter, and $\tilde{\gamma} = 1 - \gamma$.

$$\alpha^{S-N-GALM-UCB}(u,i) :=$$
$$\rho(\gamma \boldsymbol{x}_i^T \hat{\boldsymbol{\theta}}_x + \tilde{\gamma} \hat{\boldsymbol{\theta}}_z^T \boldsymbol{z}_u) + \beta(\gamma \|\boldsymbol{x}_i\|_{A_i^{-1}} + \tilde{\gamma} \|\boldsymbol{z}_u\|_{A_u^{-1}}), \tag{3}$$

where $A_i = D_i^T D_i + \boldsymbol{I}_{d_1}$, with $D_i \in \mathbb{R}^{n_i \times d_1}$ as a design matrix at iteration $t$, where each row contains item representations that user $u$ has been observed up to iteration $t$; $A_u = D_u^T D_u + \boldsymbol{I}_{d_2}$, with $D_u \in \mathbb{R}^{n_u \times d_2}$ be a design matrix at iteration $t$, where each row contains user representations that item $i$ has been recommended to up to iteration $t$. In this way, the additive model handles the user and item uncertainties separately.

**Shared Neural Generalised Bilinear UCB (S-N-GBLM-UCB)** Inspired by the Bilinear UCB algorithm (rank $r$ Oracle UCB) proposed by Jang et al. (2021), we consider a generalised bilinear model, where we assume $\mathbb{E}[y_{u,i}|\boldsymbol{x}_i, \boldsymbol{z}_u] = \rho(\boldsymbol{x}_i^T \boldsymbol{\theta}^* \boldsymbol{z}_u)$, with the coefficient $\boldsymbol{\theta}^*$ shared by all user-item pairs,

$$\alpha^{S-N-GBLM-UCB}(u,i) := \rho(\boldsymbol{x}_i^T \hat{\boldsymbol{\theta}} \boldsymbol{z}_u) + \beta \|vec(\boldsymbol{x}_i \boldsymbol{z}_u^T)\|_{W_t^{-1}}, \tag{4}$$

where $W_t = W_0 + \sum_{s=1}^{t-1} vec(\boldsymbol{x}_{i_s} \boldsymbol{z}_{u_s}^T) vec(\boldsymbol{x}_{i_s} \boldsymbol{z}_{u_s}^T)^T \in \mathbb{R}^{d_1 d_2 \times d_1 d_2}$, and $W_0 = \mathrm{I}_{d_1 d_2}$. Computing the confidence interval might be computationally costly due to the inverse of a potentially large design matrix. Different from Jang et al. (2021), instead of recommending a pair of arms $(u, i)$, we consider the item $i$ as an arm, and user $u$ as side information instead of an arm. The two-tower model in recommender systems is naturally expressed in terms of bilinear structure.

A bilinear bandit can be reinterpreted in the form of linear bandits (Jang et al., 2021), $\boldsymbol{x}_i^T \boldsymbol{\theta}^* \boldsymbol{z}_u = \langle \mathrm{vec}(\boldsymbol{x}_i \boldsymbol{z}_u^\top), \mathrm{vec}(\boldsymbol{\theta}^*) \rangle$. So linear bandits policies can be applied on bilinear bandits problem with regret upper bound $\tilde{\mathcal{O}}(\sqrt{d_1^2 d_2^2 T})$, where $\tilde{\mathcal{O}}$ ignores polylogarithmic factors in $T$. However naive linear bandit approaches cannot fully utilise the characteristics of the parameters space. The bilinear policy (Jang et al., 2021) shows the regret upper bound $\tilde{\mathcal{O}}(\sqrt{d_1 d_2 dr T})$, with $d = \max(d_1, d_2)$ and $r = \mathrm{rank}(\boldsymbol{\theta}^*)$.

Table 2: The statistical information of datasets.

|  | MIND | MovieLens-20M |
|---|---|---|
| # Users | 100,000 | 138,493 |
| # Items | 161,013 | 27,278 |
| # Topics | 285 | 20 |
| # Samples | 24,155,470 | 20,000,263 |

Table 3: Summary of UCB policies. Recall $\boldsymbol{x}_i, \boldsymbol{z}_u$ are the item and user representation, $\hat{\boldsymbol{\theta}}_u, \hat{\boldsymbol{\theta}}_x, \hat{\boldsymbol{\theta}}_z, \hat{\boldsymbol{\theta}}$ are coefficients in generalised linear models, with respect to each user, all items, all users, and all user-item pairs respectively. $f_{t-1}^u, f_{t-1}^n$ are user, item encoders up to iteration $t-1$. $\rho$ is the sigmoid function. $Y_{u,i}^n$ is a list of predictions via Monte-Carlo dropout, $\tilde{\gamma} = 1 - \gamma \in (0,1)$. Note we allow biases inside the parameters, i.e., $\boldsymbol{x}_i$ is argumented as $(x_{i,1}, \ldots, x_{i,d}, 1)$.

| Policy Name | Context | Coefficients Parameters | Predicted Rewards | Predicted Uncertainty |
|---|---|---|---|---|
| GLM | $\boldsymbol{x}_i$ | $\hat{\boldsymbol{\theta}}_u$ | $\rho(\boldsymbol{x}_i^T \hat{\boldsymbol{\theta}}_u)$ | $\|\boldsymbol{x}_i\|_{M_u^{-1}}$ |
| N-GLM | $\boldsymbol{x}_i$ | $\hat{\boldsymbol{\theta}}_u, f_{t-1}^n$ | $\rho(\boldsymbol{x}_i^T \hat{\boldsymbol{\theta}}_u)$ | $\|\boldsymbol{x}_i\|_{M_u^{-1}}$ |
| N-Dropout | $\boldsymbol{x}_i, \boldsymbol{z}_u$ | $f_{t-1}^u, f_{t-1}^n$ | $\mathrm{mean}(Y_{u,i}^n)$ | $\mathrm{std}(Y_{u,i}^n)$ |
| S-N-GALM | $\boldsymbol{x}_i, \boldsymbol{z}_u$ | $f_{t-1}^u, f_{t-1}^n, \hat{\boldsymbol{\theta}}_x, \hat{\boldsymbol{\theta}}_z$ | $\rho(\gamma \boldsymbol{x}_i^T \hat{\boldsymbol{\theta}}_x + \tilde{\gamma} \hat{\boldsymbol{\theta}}_z^T \boldsymbol{z}_u)$ | $\gamma\|\boldsymbol{x}_i\|_{A_i^{-1}} + \tilde{\gamma}\|\boldsymbol{z}_u\|_{A_u^{-1}}$ |
| S-N-GBLM | $\boldsymbol{x}_i, \boldsymbol{z}_u$ | $f_{t-1}^u, f_{t-1}^n, \hat{\boldsymbol{\theta}}$ | $\rho(\boldsymbol{x}_i^T \hat{\boldsymbol{\theta}} \boldsymbol{z}_u)$ | $\|vec(\boldsymbol{x}_i \boldsymbol{z}_u^T)\|_{W_t^{-1}}$ |

As will be shown in Section 4, S-N-GALM is computationally efficient and learn faster since the parameter space is relatively small. S-N-GBLM captures the uncertainty of user-item interaction and outperforms the additive based approach when there are more interactions (samples) in the dataset.

## 4 Experiments

**Datasets** We conduct experiments on two real-world recommendation datasets: 1) MIND [2] (Wu et al., 2020), which was collected from the user behaviour logs of Microsoft News . We used the training and validation data split as shown in Wu et al. (2020) and used the *subcategory* as our topic. 2) MovieLens-20M (Harper & Konstan, 2015), which was collected from the user ratings of movies, which is continuous values in $[0, 5]$, in MovieLens. To turn the user ratings into CTR tasks, we label samples with a rating greater than or equal to 4 as positive and the rest as negative, following (Zhou et al., 2018). As we work on adaptive personalised recommendations, we followed the data splitting strategy in (Wu et al., 2020) which splits the data by date, not by userID. Specifically, we used the user behaviour data collected from Jan 09, 1995 to Jan 01, 2010 as history logs, from Jan, 01, 2010 to Jun 01, 2014 as training data and the rest as validation data. We used the first *genres* provided in the dataset as our topics. The detailed statistics of the two datasets are shown in Table 2. Compared with Movielens-20M, MIND has sparser user interactions. For most datasets like the two datasets we used, topics are given information. For those datasets which do not provide topics, we can pre-process (e.g. clustering) items according to features to create groups as initial topics.

**Setup and Evaluation** We simulate the sequential recommendation based on the two real-world datasets. The experiments run in $T$ independent trials. For each trial $\tau \in [1, T]$, we randomly select a set of users $\mathcal{U}_\tau$ from the whole user set as the candidate user dataset from trial $\tau$. We randomly select $\epsilon\%$ of samples $\mathcal{S}_{known}$ from the training dataset as known data to the bandit models and can be used to pre-train the parameters of bandits neural model. We use the state-of-the-art two-tower model NRMS (Wu et al., 2019) for both the topic and item stage. Note we have removed the samples of the users in $\mathcal{U}_\tau$ from $\mathcal{S}_{known}$ for each trial $\tau$ to avoid information leaking from the training set to the validation and test set. This simulates the case where in recommender system we have collected some historical clicks for other users and we would

---

[2] https://msnews.github.io/

Table 4: Cumulative CTR for one stage policies with different numbers of users. We recommend 5 items for each user and simulate the experiment with 2,000 iterations, 5 trials. In policy names, "S" means shared parameters, and "N" means using neural contextual information from the NRMS.

| Dataset | MIND | | | MovieLens-20M | | |
|---|---|---|---|---|---|---|
| # User Policy | 10 | 100 | 1,000 | 10 | 100 | 1,000 |
| Random | $320 \pm 4$ | $320 \pm 2$ | $320 \pm 2$ | $242 \pm 3$ | $242 \pm 2$ | $242 \pm 2$ |
| GLM | $442 \pm 4$ | $340 \pm 4$ | $320 \pm 6$ | $459 \pm 13$ | $466 \pm 10$ | $497 \pm 8$ |
| N-GLM | $1,140 \pm 39$ | $522 \pm 58$ | $341 \pm 11$ | $1,052 \pm 61$ | $842 \pm 20$ | $552 \pm 33$ |
| N-Greedy | $1,188 \pm 72$ | $1,244 \pm 38$ | $1,282 \pm 46$ | $1,195 + 49$ | $1,140 \pm 25$ | $1,194 \pm 44$ |
| N-Dropout | $1,198 \pm 41$ | $1,256 \pm 34$ | $1,286 \pm 44$ | $1,168 \pm 49$ | $1,145 \pm 59$ | $1,164 \pm 37$ |
| S-N-GALM | $\mathbf{1,538 \pm 20}$ | $\mathbf{1,522 \pm 20}$ | $\mathbf{1,540 \pm 19}$ | $\mathbf{1,243 \pm 52}$ | $1,253 \pm 19$ | $\mathbf{1,250 \pm 40}$ |
| S-N-GBLM | $1,402 \pm 42$ | $1,366 \pm 21$ | $1,362 \pm 38$ | $1,238 \pm 49$ | $\mathbf{1,281 \pm 42}$ | $1,249 \pm 48$ |

Table 5: Cumulative CTR for policies with different number of recommendations each iteration. We select 100 users and simulate the experiment with 2,000 iterations, 5 trials. The prefix "2-" indicates two-stage policies.

| Dataset | MIND | | | MovieLens-20M | | |
|---|---|---|---|---|---|---|
| # Rec Policy | 3 | 5 | 10 | 3 | 5 | 10 |
| Random | $308 \pm 2$ | $320 \pm 2$ | $300 \pm 3$ | $249 \pm 3$ | $242 \pm 2$ | $244 \pm 2$ |
| N-Greedy | $1,013 \pm 31$ | $1,244 \pm 38$ | $1,364 \pm 32$ | $1,017 \pm 11$ | $1,140 \pm 25$ | $1,378 \pm 24$ |
| N-Dropout | $1,001 \pm 47$ | $1,256 \pm 34$ | $1,368 \pm 30$ | $1,004 \pm 33$ | $1,145 \pm 59$ | $1,328 \pm 39$ |
| S-N-GALM | $1,348 \pm 58$ | $1,522 \pm 20$ | $1,506 \pm 42$ | $1,067 \pm 71$ | $1,253 \pm 19$ | $1,453 \pm 7$ |
| S-N-GBLM | $1,211 \pm 28$ | $1,366 \pm 21$ | $1,443 \pm 6$ | $942 \pm 38$ | $1,281 \pm 42$ | $1,445 \pm 46$ |
| 2-Random | $247 \pm 2$ | $252 \pm 4$ | $250 \pm 2$ | $229 \pm 3$ | $242 \pm 1$ | $227 \pm 3$ |
| 2-N-Greedy | $1,431 \pm 37$ | $1,450 \pm 28$ | $1,315 \pm 28$ | $1,305 \pm 49$ | $1,281 \pm 36$ | $1,423 \pm 33$ |
| 2-N-Dropout | $1,462 \pm 27$ | $1,470 \pm 38$ | $1,326 \pm 28$ | $1,299 \pm 21$ | $1,358 \pm 40$ | $1,421 \pm 57$ |
| 2-S-N-GALM | $\mathbf{1,678 \pm 43}$ | $\mathbf{1,674 \pm 45}$ | $\mathbf{1,578 \pm 25}$ | $1,447 \pm 128$ | $1,403 \pm 57$ | $1,462 \pm 13$ |
| 2-S-N-GBLM | $1,608 \pm 39$ | $1,632 \pm 23$ | $1,556 \pm 22$ | $\mathbf{1,572 \pm 45}$ | $\mathbf{1,496 \pm 9}$ | $\mathbf{1,495 \pm 3}$ |

like to recommend items to new users sequentially and learn their interests. In each iteration $t$ of the total $N$ simulation iterations within each trial $\tau$, we randomly sample a user $u_t \in \mathcal{U}_\tau$ to simulate the way user $u_t$ randomly shows up to the recommender system.

We evaluate the performance by the cumulative rewards as defined in Definition 1. To make the score more comparable between different numbers of recommendations, we further define the click-through-rate (CTR) inside a batch of $m$ recommendations at iteration $t$ for each trial $\tau$ as $\text{CTR}_t^\tau = \frac{1}{m} \sum_{i=1}^m \mathbb{I}\{y_t^\tau = 1\}$. Then we evaluate the performance of bandits policies by the cumulative CTR over $N$ iterations $\sum_{t=1}^N \text{CTR}_t^\tau$. We report the mean and standard deviation of the cumulative reward or CTR over $T$ trials.

**Parameter setting** We update item neural models every 100 iterations, topic neural models every 50 iterations, topic and item generalised linear models every iteration (if there exist clicks from recommendations). We inference Monte-Carlo dropout 5 times and dropout rate is set to be 0.2. Dropout is applied in the item encoder after the word embedding layer and multi-head attention layer. Since our user representation is based on the clicked item representation, the dropout uncertainty includes both the user and item uncertainty. We set the minimum topic construction size as 1,000. We train the generalised linear models by gradient descent and select the learning rate as 0.01 (for bilinear learning rate is 0.001). Except specified, we set the UCB parameter $\beta = 0.1$.

**NRMS Item and Topic Neural Model** For items, we follow the NRMS model (Wu et al., 2019). It contains *item encoder* and *user encoder*. The item encoder learns item representation from item titles, which contains a word embedding layer, word-level multi-head self-attention and additive work attention network.

The user encoder learns user representation from their browsed/rated item, which contains item-level multi-head self-attention and additive item attention network. We follow the hyperparameter settings in (Wu et al., 2019) and change the item representation dimension to 64.

We use the same architecture and hyperparameter settings of *item encoder* and *user encoder* as in the Item Neural Model to get user representation. For each topic, we randomly initialise a vector with the same dimension of the user representation. The topic encoder takes the topic name as input and contains a word embedding layer and a multi-layer perceptron. We use dot product between the user representation and the topic representation to get user interest in topics in stage one and train the model with binary cross-entropy. To balance the positive and negative samples, we further adopt the negative sampling approach (Wu et al., 2019) with positive and negative sample ratio as 1.

**Computational Budget** To illustrate how the computational complexity of algorithms influences the performance, we follow Song et al. (2021) and introduce the *computational budget B*, which constrains the maximum number of acquisition score over arms one can compute before conducting the recommendation. The computational budget is set to evaluate the computational efficiency of algorithms and is meaningful for practical applications like large-scale recommender systems. With the same computational budget, the higher the cumulative CTR is, the more efficient the algorithm is. For one-stage algorithms, we randomly sample $B$ items from the whole item set for the candidate item set of iteration $t$; for two-stage algorithms, we first query all topics and then use the remaining budget to explore the items. The computational budget is set as $B = b \times n_{rec}$, where $b$ is the budget for each recommendation in one iteration and is set according to number of items, precisely, we set $b = 1000$ for MIND and $b = 200$ for MovieLens-20M. $n_{rec}$ is the number of recommendations, we consider $n_{rec} = \{3, 5, 10\}$ in our experiments.

**Off-Policy Bandit Simulator** Evaluation of contextual bandit algorithms on recommendation systems is challenging. On the one hand, deploying algorithms in live recommender systems can be logistically and economically expensive. On the other hand, directly evaluating the sparse recommendation data would constrain the exploration effects. We consider the *off-policy evaluation* approach, and build a *user-choice simulator* to simulate user feedback for any given items. We train the simulator on the logged data and evaluate different methods with the same simulator.

Directly learning from the logged data suffers selection bias and affects the simulator learned from it. We adopted off-policy evaluation (Huang et al., 2020; Song et al., 2021) to train a bandit simulator via logistic regression (Schnabel et al., 2016) and *Inverse Propensity Score* (IPS) (Imbens & Rubin, 2015) which re-weighs a training sample with its inverse frequency probability. We illustrate the details of our simulator training in Appendix. We then convert the predicted scores $\hat{y} \in [0, 1]$ from the simulator to binary rewards $\{0, 1\}$ by picking a threshold ($THRES = 0.384$ for MIND dataset and $THRES = 0.855$ for MovieLens-20M) in order to serve the bandits simulation, i.e. $y_{u,i} := \mathbb{I}\{\hat{y}_{u,i} \geq THRES\}$ We pick the threshold that well separates the distributions of positive and negative samples and that the positive samples are sparse enough as compared to the negative samples to reflect the real-world recommendation scenarios where the frequency of positive samples is much smaller than that of negative samples. Such threshold approach gives a deterministic binary reward. In practice, however, the user feedback can be stochastic. For example, given a fixed user and fixed item, the user might not click/rate on that item when (s)he sees the item for the first time but not for later times when (s)he sees the item again as this time his/her preference might have changed. To model such user preference uncertainty, we simply flip the value of the deterministic reward with probability $p = 0.1$. This flipping reward is our modelling choice rather than a data-driven choice as it is difficult to infer a user's preference uncertainty from a fixed dataset.

## 4.1 Main Results

Our experiment is conducted in python 3.8 (with PyTorch 1.9). We run our experiments on 2 Titan V GPUs. We first evaluate the bandits policies for one-stage exploration to illustrate the effectiveness of our proposed policies. We evaluated all policies with 2,000 iterations and 5 trials for the number of users as $\{10, 100, 1000\}$. We compare our proposed bandit policies with the following baselines, which are summarised in Table 3. We show the cumulative CTR with one standard deviation in Table 4.

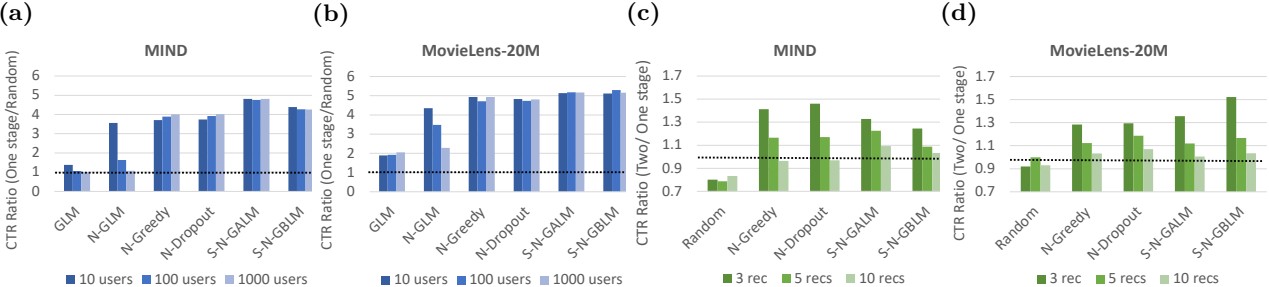

Figure 2: Cumulative CTR ratios. All policies are UCB policies except the greedy policies. (a) and (b): Improvement over random policy: the ratio of cumulative CTR of one-stage policies in Table 4 and cumulative CTR of random policy for different numbers of users. (c) and (d): Improvement of two-stage design: cumulative CTR ratio between two-stage and one-stage corresponding policies in Table 5. The dashline shows the ratio equals 1.

1. *Random*: we recommend items uniformly random from the sampled item set $\mathcal{N}_t$.

2. *Generalised Linear Model Upper Confidence Bound (GLM-UCB)* (Filippi et al., 2010): as shown in Eq. (2), the item representation uses GloVe (Pennington et al., 2014) vectors of the item titles, we learn $\hat{\boldsymbol{\theta}}_u$ with collected data for each user $u$.

3. *Neural GLM-UCB (N-GLM-UCB)*: we extend neural LinUCB (Xu et al., 2020) to neural generalised linear UCB, where we first get the deep contextual representation learnt from the NRMS model, and then follow the same acquisition function as in Eq. (2).

4. *Neural Greedy (N-Greedy)*: we recommend arms greedily with the NRMS model predictions.

5. *Neural Dropout Upper Confidence Bound (N-dropout-UCB)*: following (Gal & Ghahramani, 2016; Riquelme et al., 2018), for each user-item pair $(u, i)$, one can predict the click scores with Monte-Carlo dropout, $Y_{u,i}^n = [\hat{y}_{u,i}^1, \ldots, \hat{y}_{u,i}^n]$, where $\hat{y}_{u,i} = f_{t-1}^{user}(u)^T f_{t-1}^{item}(i)$. Then using the mean of the predictions $\bar{y}_{u,i}$ as central tendency and the standard deviation $\hat{\sigma}_{u,i}$ as uncertainty, one can follow UCB policy in Eq. (1). We infer 5 times with dropout enabled.

For all UCB based algorithm, following (Li et al., 2010; Song et al., 2021) we set the exploitation-exploration balance parameter as a fixed value ($\beta = 0.1$). For policies with neural representations, we use the NRMS model to learn user and item/topic representations and update the model parameters every 100 rounds.

For the two-stage experiments, we used 100 users, 2000 iterations and 5 trials. We tested recommendation size $\{1, 5, 10\}$ in each iteration for each user and show results in Table 5. We select the one-stage policies in Table 4 that perform well (beyond $1,000$ cumulative CTR) under 100 users, namely N-Greedy, N-DropoutUCB, S-N-GALM-UCB and S-N-GBLM-UCB, and test their performance with additional topic-stage exploration. For two-stage policies, topic and item parts follow the same policy and are denoted with the prefix "2-". For example, 2-S-N-GALM-UCB means we use S-N-GALM-UCB for topic recommendation (with NRMS topic and user encoders) and S-N-GALM-UCB for item recommendation (with NRMS item and user encoders). The last four rows follow the Algorithm 1. For two-stage Random policy, we first select topics uniformly at random from all topics and then randomly select items from the selected topics. We set the dynamic topic minimum size as the same as the computational budget.

### 4.1.1 Observations and Interpretations

We discuss the observations and interpretations based on the results in Table 4 and 5. We further illustrate the CTR ratios in Figure 2.

**Exploration improves CTR:** We can observe in Table 4 that a proper level of exploration during adaptive recommendation can improve the cumulative CTR. Our proposed policies balance the exploitation and

exploration with a new way of computing uncertainties and making use of neural representation, which returns higher CTR scores compared to N-Greedy (no exploration). This suggests an informative uncertainty based exploration can alleviate the feedback loop bias and learn user interest dynamically and adaptively.

**Proposed bandit policies outperform others:** Both of our proposed policies (S-N-GBLM-UCB, S-N-GALM-UCB) have higher cumulative CTR compared to other polices in Table 4 and 5, which illustrates the effectiveness of the shared model and usage of the user representation from neural network with additive (S-N-GALM-UCB) and bilinear (S-N-GBLM-UCB) structure. The improvement on the MIND dataset is larger than on the Movielens-20M dataset, since our shared parameter and interaction design can capture the sparse interactions better than the baseline policies. Our results verify the two design motivations of our proposed policies:

- **Neural representation improves the Performance:** In Table 4, compared with non-neural policies (first two rows), the neural network based policies (last 5 rows) has significant improvement. We further illustrate this in Figure 2a and 2b, which show the cumulative CTR ratio between one-stage policies and random policies. Particularly, N-GLM-UCB outperforms GLM-UCB, where the only difference is that N-GLM-UCB makes use of neural item representation from the two-tower model (with regular updating) while GLM-UCB uses GLoVe directly. Our proposed policies further improves the cumulative CTR by making use of neural user and item representations from two-tower model in an additive or bilinear way. This shows the power of combining two-tower neural representations into the bandit recommendation framework.

- **Shared weights for bandit model improves CTR:** In Table 4, we can see the cumulative CTR for the disjoint policies like GLM-UCB and N-GLM-UCB drops dramatically when the number of users increases (i.e., the number of samples per user decreases), which shows the disjoint models are hard to be scalable to the large user or item recommender system. This is because the disjoint policies need enough samples to learn the coefficients for each user. Our proposed policies, which extend from N-GLM-UCB to share parameters across different users or items, outperform disjoint policies.

**Two-stage algorithms are efficient:** Recall that we set the same computational budget for one-stage and two-stage algorithms to constrain the number of scores we can calculate. The higher the cumulative CTR is, the more efficient the algorithm is. In Table 5, two-stage policies outperform corresponding one-stage policies since the topic exploration scope in the item space under promising topics and save the computational budget. The exception is the two-stage Random policy, which is worse than one stage Random since selecting bad topics at the first stage would limit the item selection and lead to a lower click rate. This further shows the importance of a reasonable topic recommendation. We visualise the cumulative CTR ratio between two-stage and one-stage policies in Figure 2c and 2d, where we can see except the random policy the ratios are above 1. We further show an ablation study of the CTR and computational time when we do not constrain the computational budget in Table 6.

### 4.2 Ablation Study

**Dynamic topic clustering** We test how our dynamic topic construction in Algorithm 2 influences the performance of two-stage policies in Table 5. We set the minimum reconstruct size $p = 1000$ for MIND and $p = 200$ for MovieLens-20M. We compare Algorithm 1 (denoted as *Dynamic Topic* in Figure 3) with *No Dynamic Topic* case, i.e., remove line 8 in Algorithm 1 and only the top topic is recommended. To make the comparison fair, if the number of candidate items under the recommended topic is smaller than the computational budget, we randomly sample items from the whole item set to guarantee the numbers of items evaluated for the two methods are the same. Figure 3 shows that for the tested policies, dynamically forming topics improves the CTR significantly.

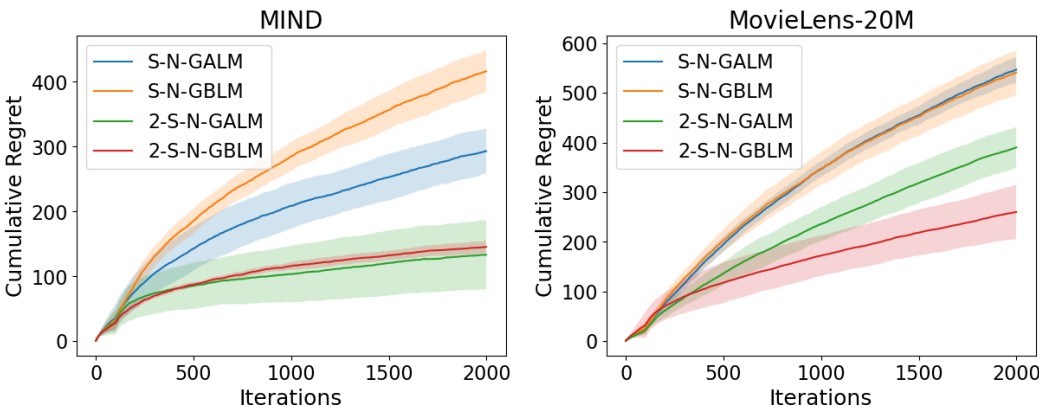

Figure 4: Cumulative regret for our proposed algorithms (2,000 iterations, 5 trials, 5 recommendations and 100 users). We report one standard deviation. This verifies that our algorithms empirically obtain the sublinear regret.

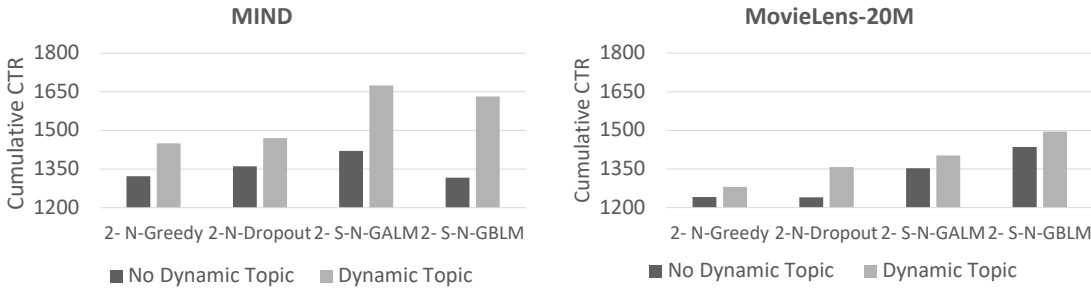

Figure 3: The effect of dynamic topics forming (2,000 iterations, recommendation size 5 and user size 100.).

**Without budget constraint**   To further illustrate how our two-stage framework addresses the computational complexity for large item space, we conduct an additional experiment without the computation budget constraints. We list the cumulative CTR and running time for recommendation stage in each iteration for our proposed policies with one-stage and two-stage recommendation in Table 6. We can observe that for the same policy, first recommending topics (two-stage) is much faster than recommending from the whole item space as we analysed in Section 3.1, while the cumulative CTR remains similar for one-stage and two-stage. The running time gap is larger when the number of items is larger (e.g., in MIND). When the item space is much larger than the topic space, directly recommending items from the whole item space has the benefits of scanning through all item scores and tends to have higher CTR (e.g., in MIND), with the cost of longer running time. However, long loading time in large commercial recommender systems may influence the user experience and lose potential long-term users.

**Cumulative Regret Curve**   Given our simulator design, we assume in each round, with flipping probability 0.1, the expected optimal reward in each iteration is $1 \times (1 - 0.1) = 0.9$. The cumulative regret then can be calculated via Definition 1. We show the cumulative regret curve in Figure 4, which illustrates how our algorithm performs with respect to the iterations and verifies the sublinear regret empirically.

## 5   Related Work

**Large Space Exploration**   To address large item spaces, hierarchical search is employed. For two-stage bandits work, Hron et al. (2021) studied the effect of exploration in both two stages with linear bandit

Table 6: Cumulative CTR and computational time for recommendation when there is no computation budget limit (2,000 iterations, recommendation size 5 and user size 100). Dynamic topic minimum size is set as # items/ # topics.

| Dataset | MIND | | MovieLens-20M | |
|---|---|---|---|---|
| Metric
Policy | CTR | Time (s) | CTR | Time (s) |
| S-N-GALM | $1,739 \pm 35$ | $1.31 \pm 0.13$ | $1,595 \pm 301$ | $0.36 \pm 0.05$ |
| S-N-GBLM | $1,674 \pm 26$ | $16.70 \pm 0.24$ | $1,668 \pm 145$ | $3.98 \pm 0.28$ |
| 2- S-N-GALM | $1,673 \pm 39$ | $0.17 \pm 0.04$ | $1,695 \pm 19$ | $0.12 \pm 0.03$ |
| 2- S-N-GBLM | $1,650 \pm 6$ | $0.88 \pm 0.34$ | $1,732 \pm 15$ | $0.87 \pm 0.17$ |

algorithms and Mixture-of-Experts nominators. Ma et al. (2020) proposed off-policy policy-gradient two-stage approaches, however, without explicit two-stage exploration. There is also a branch of related work considering hierarchical exploration. Wang et al. (2018); Song et al. (2021) explored on a pre-constructed tree of items in MAB or linear bandits setting. Zhang et al. (2020) utilises key-terms to organise items into subsets and relies on occasional conversational feedback from users. Hong et al. (2022a) proposed a hierarchical Thompson sampling algorithm in parameter space with Gaussian hierarchies implementation, and Hong et al. (2022b); Aouali et al. (2022) extended it into multi-level hierarchies and mixed effect (categories). To the best of our knowledge, no existing work has studied two-stage exploration for deep contextual bandits.

**Neural Contextual Bandits**  Contextual bandits with deep models have been used as a popular approach since it utilises good representations. Riquelme et al. (2018) conducted a comprehensive experiment on deep contextual bandit algorithms based on Thompson sampling, including dropout, neural-linear and bootstrapped methods. Recently, deep contextual bandits have been applied to recommender systems. Collier & Llorens (2018) proposed a Thompson sampling algorithm based on inference time Concrete Dropout (Gal et al., 2017) with learnable dropout rate, and applied this approach to marketing optimisation problems at HubSpot. Guo et al. (2020) studied deep Bayesian bandits with a bootstrapped model with multiple heads and dropout units, which was evaluated offline and online in Twitter's ad recommendation. Hao et al. (2022) added representation uncertainty for embedding to encourage further exploring items whose embeddings have not been sufficiently learned based on recurrent neural networks. Zhu et al. (2022) proposed practical algorithms for linearly constructed large action space with the inverse gap weighting method. Duran-Martin et al. (2022) proposed Bayesian neural networks based methods with constant memory for contextual bandits.

Theoretically, Zhou et al. (2020) proposed NeuralUCB and proved a sublinear regret bound, followed which Gu et al. (2021) studied the case where the parameters of DNN only update at the end of batches. Xu et al. (2020) proposed Neural-LinUCB to make the use of deep representation from deep neural networks and shallow exploration with a linear UCB model, and provided a sublinear regret bound. Zhu & Rigotti (2021) proposed sample average uncertainty frequentist exploration, which only depends on value predictions on each action and is computationally efficient. Kassraie & Krause (2022) studied regret bounds for the neural contextual bandits for fully-connected and convolutional networks.

To the best of our knowledge, among those utilising the power of deep representation from existing network structures in online recommender systems with bandits feedback, no existing work has applied two-stage dynamic exploration, which increases computational efficiency and is important to large-scale recommender systems in practice. And we are the first work to utilise the generalised additive and bilinear model for deep neural network based exploration, which suits the two-tower recommender system naturally. Therefore, the related work mentioned in this section considers different settings and cannot be directly compared with our methods.

## 6 Conclusion and Future Work

We consider the personalised recommendation task as a sequential decision making problem, and use contextual bandits to balance exploitation and exploration. To increase computational efficiency in large arm spaces, we propose a two-stage topic-item recommendation framework with dynamically generated topic groups. We utilise the deep representation from a two-tower model and propose the generalised additive and bilinear upper confidence bound (S-N-GALM, S-N-GBLM) policies to generate recommendations. Empirical experiments on two large-scale recommendation datasets (MIND, MovieLens-20M) show our proposed two-stage framework is computationally efficient and our proposed policies outperform baselines. Our work addresses the feedback-loop bias and computational efficiency for large-scale online recommender systems, which help practitioners improve the recommendation quality and speed. Although there are theoretical analyses for simplified settings of hierarchical designs (Hong et al., 2022b; Aouali et al., 2022) and neural contextual bandits (Zhou et al., 2020; Kassraie & Krause, 2022), the choice of dynamic topics and deep neural networks bring extra challenges on full theoretical analysis to our framework, we will leave it to future work. To the best of our knowledge, our work is the first to consider two-stage dynamic deep efficient exploration for large-scale recommender systems. Our work contributes to the empirical efficient usage of contextual bandits approaches for large-scale recommendation communities.

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
