# OpenReview forum: "Two-Stage Neural Contextual Bandits for Adaptive Personalised Recommendations"
_TMLR — Rejected by TMLR_

### Review · Reviewer_Qe7o · 2023-04-11

**Summary Of Contributions:**

In this paper, the authors design a recommender system (RS) for an online scenario: at each iteration, the RS is given a user and a set of allowed items (accompanied by some features) and has to recommend $m$ of these items. Afterwards, the RS collects a feedback per item, which it uses for the next recommendations. The authors also impose a budget on the number of scores the RS is allowed to select from the set of $m$ items.

The proposed RS uses a 2-step process to derive the recommendation: first, it selects $m$ promising topics, and then it restricts the recommendation to items with these topics. The experiments derived from the MIND and MovieLens-20M datasets demonstrate the interest of the 2-stage process.


**Audience:**

Yes

**Claims And Evidence:**

No

**Requested Changes:**

# To clarify the need for personalisation
1. Add the results of GLM with a unique vector-parameter $\theta_x$ (to compare with the results of the current version of GLM).
2. Find a scenario with a clear effect of personalisation (it may require a different dataset and/or more iterations per trial so that the RS has enough time to learn personalised recommendations).

# To clarify the need for exploration
1. In Figure 4, the addition of the curve of GLM-Greedy would allow to check if it is able to learn (due to the "exploration" enforced by the setting). For a more detailed discussion, it would also be useful to add GLM-Greedy without budget constraint (to check whether this constraint is the source of exploration or not) and to add the curves of GLM and GLM-Dropout (to check whether they learn more than GLM-Greedy).
2. Similarly, adding greedy flavors of S-N-G.LM algorithms would clarify the need for exploration (or not).
3. Find a scenario with a clear need for exploration (it may require a different data set and/or more iterations per trial so that the RS has enough time to benefit from exploration).

# To clarify the use of datasets
Paragraph "Datasets" mentions a split of the datasets into three parts, but in the rest of the paper it is unclear which part is used for what. Typically, what data is used to train the bandit simulator? Or what is the "test set" mentioned in Paragraph "Setup and Evaluation"?

# (Little) surprises in the text
Some of the information given in Paragraph parameter setting" is difficult to understand, as it refers to objects that have not been presented or warned beforehand: the Monte-Carlo bandit, the presence of attention layers in the item-feature network.

Similarly, Paragraph "datasets" should warn against the use of a bandit simulator (to be presented later).



**Strengths And Weaknesses:**

# Strength: the 2-stages process increases the reward collected

# Weakness: topics must be known in advance
The proposed approach relies heavily on topics that are both known in advance and correlated with users' tastes. The authors mention that such topics could be extracted by clustering, but such a change has to be done with care, as the clustering step will both increase the computational budget and make the clusters used by the first stage of the recommendation evolve over iterations.

# Weakness: two aspects of the recommendation scenario considered in the paper are not met in the experiments (the need for personalization and the need for exploration).
First, in the experiments, S-N-GALM and 2-s-N-GALM are the best approaches on the MIND dataset and often the best approaches on MovieLens. Since these algorithms are the ones that do not personalize their recommendation, it raises questions about the experimental setting to evaluate recommender systems.

Secondly, if we restrict the discussion to N-GLM, N-Greedy and N-Dropout, N-Greedy and N-Dropout perform similarly well. This suggests that with the current experimental setup, greedy algorithms are sufficient. This could be because the number of iterations is too small, or because the allowed items are randomly selected (to enforce the budget constraint), which forces a kind of exploration.


# Weakness: Reproducibility  of the experiments is hindered by an unclear presentation of the use of the datasets

---

> ### Author Response · Authors · 2023-05-10
>
> **Weakness**: topics must be known in advance
> **Response**: The clustering step if needed, can be a pre-processing step and does not need to run again through the experiment. Since we will have item representations, which can be used for the clustering step.
>
> **Weakness**: two aspects of the recommendation scenario considered in the paper are not met in the experiments (the need for personalization and the need for exploration).
> **Response**: S-N-GALM and 2-S-N-GALM do personalised recommendations, by learning and using user representations.
> N-Dropout adopted a simple test-time dropout samples method to estimate the uncertainty, which can be insufficient depending on the neural structure and sample number. That a single exploration based policy is not clearly better than greedy does NOT inform that ALL exploration policies are not useful. Instead, as clearly shown in our result Tables 4 and 5, our proposed policies, which is an exploration-based policy, are significantly better than the greedy policies. We believe this is because our policy can estimate the uncertainty better than N-dropout.
>
> **Weakness**: Reproducibility of the experiments is hindered by an unclear presentation of the use of the datasets
> **Response**: Without more detail about what the reviewer found unclear in our presentation, we unfortunately cannot improve our paper. If the reviewer meant the below request changes for dataset, we have clarified the usage.
>
> **Requested Changes**:
> To clarify the need for personalisation
> Add the results of GLM with a unique vector-parameter
>  (to compare with the results of the current version of GLM).
> Find a scenario with a clear effect of personalisation (it may require a different dataset and/or more iterations per trial so that the RS has enough time to learn personalised recommendations).
> **Response**: It is unclear what the reviewer means by “GLM with a unique vector-parameter”. Can the reviewer explain more?
> In terms of personalisation, our policies make use of user representations and keep updating them. Our contributions are mainly about computational efficiency and incorporating neural representations into bandit policies. The effect of personalisation in recommender systems has been illustrated and verified by many previous works, the reviewer can refer to Wu et al., 2020.
>
> To clarify the need for exploration
> In Figure 4, the addition of the curve of GLM-Greedy would allow to check if it is able to learn (due to the "exploration" enforced by the setting). For a more detailed discussion, it would also be useful to add GLM-Greedy without budget constraint (to check whether this constraint is the source of exploration or not) and to add the curves of GLM and GLM-Dropout (to check whether they learn more than GLM-Greedy).
> Similarly, adding greedy flavors of S-N-G.LM algorithms would clarify the need for exploration (or not).
> Find a scenario with a clear need for exploration (it may require a different data set and/or more iterations per trial so that the RS has enough time to benefit from exploration).
> **Response**:
> It is not clear what the reviewer means by “GLM-Greedy”, “GLM-Dropout”. GLM or our models in this paper is designed for calculating uncertainties (differently from the dropout approach). The effect of exploration is shown by comparing our policies to N-Greedy, which shows a clear improvement.
>
> To clarify the use of datasets
> Paragraph "Datasets" mentions a split of the datasets into three parts, but in the rest of the paper it is unclear which part is used for what. Typically, what data is used to train the bandit simulator? Or what is the "test set" mentioned in Paragraph "Setup and Evaluation"?
>
> **Response**:
> In the Dataset paragraph, we stated we split datasets into training and validation datasets: “We used the training and validation data split as shown in Wu et al. (2020) and used the subcategory as our topic.” “we used the user behaviour data collected from … as training data and the rest as validation data.”
> The simulator is trained on the training dataset (we have updated text to be clear about it), and for CB models: “. We randomly select 20% of samples S known from the training dataset as known data to the bandit models and can be used to pre-train the parameters of bandits neural model. ”
> In the Paragraph "Setup and Evaluation", we have updated to “avoid information leaking from the training set to the validation set”.
>
> (Little) surprises in the text
> Some of the information given in the Paragraph parameter setting" is difficult to understand, as it refers to objects that have not been presented or warned beforehand: the Monte-Carlo bandit, the presence of attention layers in the item-feature network.
> Similarly, Paragraph "datasets" should warn against the use of a bandit simulator (to be presented later).
> **Response**:
> Thanks for the suggestion. We have modified the text to fix it.

---

### Review · Reviewer_R3bg · 2023-04-16

**Summary Of Contributions:**

The paper considers a bandit approach to recommendation systems. A two level hierarchical structure is proposed, selecting first a set of topics, and then the set of items. Content based neural network is used for both combined with a UCB style bonuses to aid the exploration. For the item selection phase a dynamical rebalancing is also considered, mostly for computational reasons.

The proposed algorithms are evaluated empirically on two datasets, MIND and MovieLens-20M, adapted to bandit evaluation using an off-policy simulator. Te experimental results show that the algorithms have better click-through-rate compared to some alternatives considered. The dynamical topic selection approach also has a strong performance when the computational budget is limited.


**Audience:**

No

**Claims And Evidence:**

No

**Requested Changes:**

The major issue is the extent of the experiments. Longer experiments with more trails are needed to substantiate any if the claims. I am not sure if there are claims regarding the need for exploration, if it is the case, then more extensive experiments targeted to the issue are necessary. Currently the paper supports a claim that addressing computational limitation is useful, but this is a trivial claim. If the claim is regarding the proposed dynamical topic approach, then alternatives approaches to tackle large number of items should be investigated more extensively.

There are some small issues as well, like referencing twice the Appendix, which does not exist. In one of the case the referenced Table is in the text, although the list of symbols is not complete. There is some detail on the training of the simulator, but I do not know if it was intended to be more detailed.


**Strengths And Weaknesses:**

The paper seems to have two contributions: the proposed recommendation algorithms and the experimental evaluations.

Bandit approaches to recommendation systems are mostly interesting because they have a theoretical underpinning, which is not the case for this submission. Exploration in practical recommendation systems is rarely handled by bandit approaches, and I am not sure that the submission is making any attempt to analyze the effect of (and need of) exploration beyond having a baseline that is greedy. More extensive experiments on the exploration issue would be needed to provide any insights.

Addressing computational issues through dynamical topic selection could be interesting, but there is little comparison with alternative approaches to tackle large number of items (used in practical systems), and the experiments use only the static variant as baseline.

Evaluating bandit approaches is difficult in recommendation systems, and there is no established method for it. Using a simulator is one of the possible approaches, but it introduces the bias of the simulator, and forces a typically non-stationary problem (due to changing preferences of users and varying popularity and context of the items) to a stationary one. Given the pretraining phase (where the system is not evaluated yet), the need of exploration seems a marginal one on this setup.

The datasets used in the experiments are reasonable in size (there are large ones nowadays, but they are not small). However, limiting the number of iterations to 2000 is strange, and I am not even sure what is the interest in having experiments with 10 users. Averaging the experiments over 5 trials prevents any attempt to make claim of statistical significance. In summary, for an empirical paper the experiments are small. I am not sure what is the reasons for this, maybe the neural networks are too slow, but then the proposed algorithms are not really applicable in practice. The experiments are not extensive in terms of baselines either, the baselines being (dumbed down) versions of the proposed algorithms.

---

> ### Author Response · Authors · 2023-05-10
>
> **Question**: Bandit approaches to recommendation systems are mostly interesting because they have a theoretical underpinning, which is not the case for this submission. ...
> **Response**: 1) Although theoretical contributions of bandit work are of interest, there are quite a lot of bandit works focusing on empirical evaluations in recommender systems and many industries use the bandit approach for real experiments and products as listed in Section 5 Neural Contextual Bandits first paragraph. There is a gap between theoretical and empirical approaches in bandit approaches. Lots of theoretical work does not work well in empirical large-scale systems where theoretical assumptions do not hold or are computationally inefficient. We’d like to argue that building a computationally efficient pipeline for practical recommender systems is important and the theoretical analysis of our framework will be left as future work. 2) To illustrate the effect of our proposed algorithm and the need for exploration, we compared our proposed exploration strategies as the baselines (e.g. GLM, N-GLM, N-Droupout) and show results in Tables 4 and 5. So our analysis covers not only greedy strategies but also exploration baselines.
>
> **Question**: ... there is little comparison with alternative approaches to tackle large number of items (used in practical systems), and the experiments use only the static variant as baseline.
> **Response**: In the Ablation study, we compared the dynamic topic clustering with no dynamic case and show results in Figure 3, which indicates our dynamic topic construction increases the performance by a large amount.
>
> **Question**: ...Given the pretraining phase (where the system is not evaluated yet), the need of exploration seems a marginal one on this setup.
> **Response**: The simulator is trained on the whole training dataset (longer time interval), where representations used for bandit policies are pretrained on a small proportion of the training dataset, as stated in “We randomly select 20% of samples S known from the training dataset as known data to the bandit models and can be used to pre-train the parameters of bandits neural model.”.
> This allows the simulator to be able to learn and predict more extensive user interests. And bandit policies will learn through exploration with the feedback given by the simulator for the preferences not indicated in the data used for pre-training. We agree that the bandit approaches evaluation is difficult on recommender systems and is still an active research area. We follow the previous work (Huang et al., 2020; Song et al., 2021 to debias the via Inverse Propensity Score (IPS) approach.
>
> **Question**: ...for an empirical paper the experiments are small....
> **Response**: 1) We select 10,100,1000 as the number of users as a comparison of how different policies respond to the increase of the number of users in the system. As stated in Section 4.1: “In Table 4, we can see the cumulative CTR for the disjoint policies like GLM-UCB and N-GLM-UCB drops dramatically when the number of users increases (i.e., the number of samples per user decreases), which shows the disjoint models are hard to be scalable to the large user or item recommender system.
> This is because the disjoint policies need enough samples to learn the coefficients for each user.”
> 2) The number of iterations is designed based on the number of users. In our case, we have 200,20,2 samples for each user on average for the three #User settings, where 200 is large enough for the agent to learn user preference even if there are no correlations between users, while 2 is relatively small and the agent needs to learn across similar users. For a simulation, we argue the selection of the number of iterations and users is enough to show the capacity of our model and illustrate the benefits.
> 3) We select several competitive baselines, e.g. GLM, N-GLM, N-Greedy, and N-Dropout as a comparison of our algorithms. They are not just versions of our proposed algorithms.
>
> **Requested Changes**: The major issue is the extent of the experiments...
> **Response**: We have responded in terms of concrete questions about the extent of the experiments above. Our claims are clearly stated in the contributions and experiments parts. We respectfully disagree that improving computational efficiency is a trivial claim, which is an important question and we propose novel ways to increase computational efficiency with empirical evidence.
>
> **Question**: There are some small issues as well, like referencing twice the Appendix, which does not exist. In one of the case the referenced Table is in the text, although the list of symbols is not complete. There is some detail on the training of the simulator, but I do not know if it was intended to be more detailed.
> **Response**: The Appendix was submitted as supplementary material, where we put more details about the simulator. The main notations are put in Table 1 in the main paper.

---

### Review · Reviewer_dMKj · 2023-04-27

**Summary Of Contributions:**

The paper proposes a hierarchical contextual bandit framework for recommendation systems to address closed loop effects where the recommendation system gets biased with time towards a fixed set of popular items. The hierarchical contextual bandit uses predefined topics in the dataset to alleviate the challenges of exploration where the number of items are very large. Furthermore, the authors propose two improvements: (1) a way to dynamically balance the topics with respect to the number of items (2) neural additive and bilinear bandit architectures. Both of these improvements show improved results in offline simulation with respect to baselines.

**Audience:**

Yes

**Broader Impact Concerns:**

Since this work is generic and performs experiments using standard datasets, I do not see any immediate risk of significant harm.

**Claims And Evidence:**

No

**Requested Changes:**

1. The issues raised in weakness 1, 2 and 3 should be addressed.
2. There should be a deliberate discussion between the contributions in this paper and the work of Song et al 2021.



**Strengths And Weaknesses:**

## Strengths
1. The motivations for the framework are clear and this addresses important problems in recommendation systems. The experiments are well aligned with the motivations behind this work.
2. The paper shows further evidence that even simple balancing of topics in the hierarchical bandit solutions may lead to an improved performance compared to imbalance topics.
3. While both neural bandits and bi-linear bandits have been studied in prior works, the paper empirically demonstrates that applying them in a two-step exploration framework may lead to improved performance.

## Weaknesses
1. The authors claim that the paper’s contributions are (1) a two-stage hierarchical contextual bandit framework (2) balancing of topics (3) bilinear bandit structures. It is not clear how claims (1) and (2) differ from the contributions in Song et al 2021. Regarding claim (3), bilinear bandits have already been studied in Jang et al, 2021 and this may not be a novel architecture.
2. The paper claims to alleviate the closed loop effects of recommendation systems. It’s known that UCB algorithms can be concentrated towards greedy selection of arms with time. There is no empirical evidence that the proposed framework recommends a diverse set of items to a user to address the closed loop effects
3. The claim that dynamic balancing of topics improves performance is proved by the ablation study where the items from the dynamic set of topics are compared to the items from the top topic only. I think this setup proves that a diverse set of topics may be better than a single top topic and it doesn’t show that balancing of topics is the reason behind the improvement. A more rigorous evaluation should be performed where the dynamic topics should be compared to top m topics i.e line 8 in Algorithm 1 can be replaced by greedily selecting top m topics as a baseline.
4. In Table-3, both user and item representations are inputs for S–GBLM and S-GALM while N-GLM has only the item representations as input. It’s expected that S-GBLM and S-GALM would outperform the baselines with access to more features. This has not been addressed in the paper.
5. The authors claim that policies with exploration return higher CTR compared to greedy policies since bandit policies show higher CTR compared to the N-Greedy model. Why does the exploration policy show higher CTR if the policy is trained using logged training dataset only? Is this claim true given that in both Tables 4 and 5, N-Dropout shows comparable or even slightly worse performance compared to N-Greedy.

---

> ### Author Response · Authors · 2023-05-10
>
> **Question**: The authors claim that the paper’s contributions are (1) a two-stage hierarchical contextual bandit framework (2) balancing of topics (3) bilinear bandit structures. It is not clear how claims (1) and (2) differ from the contributions in Song et al 2021.
> **Response**: The main differences between our work and Song et al 2021 are: 1) The hierarchical exploration is different: in pHCB in Song et al 2021, the receptive field is constructed in terms of a fixed tree structure with top-down expansion, while we dynamically reform the topics in a down-top manner in each round according to the bandit scores. 2) How to form receptive fields/topics are different: Song et al 2021 forms the receptive fields by expanding according to the pre-fixed tree structure, while we construct the new topics via the dynamic UCB scores calculated in each round. 3) Our framework allows cooperating neural representation and updates. We added the comparison in Section 3.2.
>
> **Question**: Regarding claim (3), bilinear bandits have already been studied in Jang et al, 2021 and this may not be a novel architecture.
> **Response**: We extend the bilinear bandits structure in Jang et al 2021 to be able to use and update neural representations, with extensive empirical evaluations on real-world applications. This is important for large-scale recommender systems.
>
> **Question**: ..There is no empirical evidence that the proposed framework recommends a diverse set of items to a user to address the closed loop effects
> **Response**: 1) We aim to alleviate closed-loop effects, which are evaluated in terms of cumulative rewards/regrets. Diversity is not our direct goal: when we explore the user interests, we do not directly design a diversity recommendation set, but recommend items that the agent is uncertain about the user’s preference. 2) UCB tends to be greedy with time, only after the algorithm has done enough exploration. It is different from conducting a greedy recommendation at the beginning, which is also clearly shown in our experiments that our proposed algorithms perform better than greedy ones. And UCB types of algorithms will always allocate some level of exploration, even when iterations are large (the exploration rate would be quite small), to ensure the algorithm does not get stuck in a sub-optimal choice.
>
> **Question**: The claim that dynamic balancing of topics improves performance is proved by the ablation study where the items from the dynamic set of topics are compared to the items from the top topic only....
> **Response**: 1) In the Ablation study, we selected the top topic additionally with random items to make the size of potential items the same. 2) Greedily selecting top m topics is shown in Table 5 as the 2-N-Greedy policy. And we can see from Table 5, both of our approaches outperform the top m greedy policy.
>
> **Question**: ...It’s expected that S-GBLM and S-GALM would outperform the baselines with access to more features. This has not been addressed in the paper.
> **Response**: We clearly state how we used the representations in Section 4.1 when we outline the baselines. The new policies are designed to make use of user representations in a methodology way.
>
> **Question**: The authors claim that policies with exploration return higher CTR compared to greedy policies since bandit policies show higher CTR compared to the N-Greedy model. Why does the exploration policy show higher CTR if the policy is trained using logged training dataset only? Is this claim true given that in both Tables 4 and 5, N-Dropout shows comparable or even slightly worse performance compared to N-Greedy.
> **Response**: 1) As stated in Section 4 Off-Policy Bandit Simulator paragraph, we build a simulator based on the logged training dataset and debiased using the Inverse Propensity Score4 (IPS). So the policy is NOT “trained using logged training dataset only”, but uses the simulated rewards from our simulator. Note that the CB models are pre-trained only on 20% of the training data (as mentioned in the setup paragraph in Section 4), so the simulator knows more user preferences than CB models. 2) N-Dropout adopted a simple test-time dropout samples method to estimate the uncertainty, which can be insufficient depending on the neural structure and sample number. While a single exploration-based policy may not be clearly better than greedy, this does NOT imply that ALL exploration policies are not useful. Instead, as clearly shown in our results in Tables 4 and 5, our proposed policies, which are exploration-based policies, are significantly better than the greedy policy. We believe this is because our policy can estimate the uncertainty better than N-dropout.
>
> **Requested Changes**:
> The issues raised in weakness 1, 2 and 3 should be addressed.
> There should be a deliberate discussion between the contributions in this paper and the work of Song et al 2021.
> **Response**: Please see above for detailed responses.

---

### Decision · Action_Editors · 2023-06-16

**Recommendation:** Reject

**Comment:**

We had a brief discussion with the reviewers following the author's response. While the responses did settle some of their questions their main remarks remained. In the end, the three independent reviewers were unanimous that while this paper is clearly relevant for TMLR some of its claims are not fully supported experimentally.

Each reviewer came to this assessment for slightly different reasons. Reviewer dMKj directly outlines that some of the contributions of the paper would require to be better detailed. They also mention the limit of performing experiments for a maximum of 1,000 users for example to claim larger-scale/practical gains. Reviewer Qe7o, re-states some of the arguments from their original review and also that the evaluation is only empirical. For TMLR, I discounted the latter as far as it relates to significance. Reviewer R3bg, outlines (again) the limitations of the current empirical evaluation.

I understand that the authors replied to each comment from the reviews fairly precisely and that this outcome might be frustrating for the authors given the lack of more in-depth discussion with the reviewers. If the paper does indeed support all of its claims, then the main issue might be one of clarity with respect to the goals of the paper and the results as they are reported and discussed in the experiments.

I also realize that unfortunately, follow-up questions from the authors were not answered by one of the reviewers. The reviewer in question assured me that they had carefully considered your responses and went back to the paper for details.

I would happily accept to review a revised version of this manuscript.

**Audience:**

The topic and methods developed in this paper would be of clear interest to part of the TMLR audience.

**Claims And Evidence:**

The reviewers are of the opinion that the claims are not fully supported by the provided empirical evidence.

**Resubmission Of Major Revision:**

The authors may consider submitting a major revision at a later time.